# Investigation on Microplastics in Soil near Landfills in the Republic of Korea

Won-Kyu Kim [1], Hanbai Park [2], Kazuei Ishii [3] and Geun-Yong Ham [3,*]

1   Department of Civil and Environmental Engineering, 119 Academy-ro, Yeonsu-gu, Incheon National University, Incheon 22012, Republic of Korea; yacoortman@gmail.com
2   Korea Microplastic Research Center Co., Ltd., 2nd Floor, 88, Daejeo-Ro 299 St., Busan 46702, Republic of Korea; kmpr@kmpr.co.kr
3   Laboratory of Sustainable Material Cycle Systems, Faculty of Engineering, Hokkaido University, Kita 13, Nishi 8, Kita-ku, Sapporo 060-8628, Hokkaido, Japan; k-ishii@eng.hokudai.ac.jp
*   Correspondence: geun-yong.ham@eng.hokudai.ac.jp

**Abstract:** Microplastics can cause physical, chemical, biological, and structural problems in soil. In this study, microplastics were identified in the soil near two landfills where contamination by microplastics was expected. Pretreatment was performed to remove organic matter and to separate microplastics from the soil samples. FT-IR microscope analysis was performed to confirm the quantity and types of microplastics. The colors and shapes of microplastics in the soil were analyzed using a digital microscope. Averages of 73.4 MPs(ea)/kg and 97.8 MPs(ea)/kg of microplastics were identified in the soil at the two landfills. The main shapes of microplastics were fragments, fibers, and films, and it was confirmed that secondary plastics were found at a high rate. The major plastic types were identified as PP and PE, ranging from 62.5 to 65.3% in proportion, followed by PET, PS, nylon, PMMA, and PVC. As for the colors of microplastics, black had the highest percentage, while other microplastics were identified as being white, blue, transparent, gray, green, red, and yellow. These results can be taken as important data indicating that microplastics in the soil around landfills can be affected by landfill waste.

**Keywords:** microplastics; soil; landfill; FT-IR; shape; color; composition





## 1. Introduction

Microplastics, called MPs, are small pieces of plastics: organic synthetic polymer compounds with particles of 5 mm or less. It is difficult to identify MPs with the naked eye. Depending on the source, MPs can be divided into two types: primary, which are tiny particles designed for commercial use, for example, cosmetics, beads, and microfibers, and secondary MPs, which are particles fragmented from larger plastic items such as straws or PET bottles. MPs are a critical threat to terrestrial and aquatic ecosystems and can reach humans through the food supply chain [1–4]. The use of plastics has increased rapidly since the 1960s, and production increased from 335 million tons in 2016 to 368 million tons in 2019 [5]. Although plastic production slightly decreased to 367 million tons in 2020 due to the impact of COVID-19, it is expected to continue to increase [5]. Waste plastics are disposed of through landfills, incineration, and recycling, but untreated plastics, in the form of MPs, are expected to accumulate as they move through environmental media such as water, soil, and air. Phthalate and bisphenol A, contained in plastic as additives, are endocrine disruptors that cause genotoxicity and sexual dysfunction [6]. In particular, MPs with a large surface area can adsorb endocrine disruptors, non-degradable organic substances, and heavy metals, meaning that the risk associated with MPs is increasing.

Research on MPs has mainly been conducted in the marine environment, and there is still insufficient research on MPs in the terrestrial environment, particularly in soil conditions. The study of MPs in soil began by examining the impact of plastic pollution

caused by littering, plastic waste dumping, and the improper management of landfills. Geyer et al. (2017) reported that only 21% of plastics were recycled by reuse and incineration, whereas the remaining 79% were landfilled or dumped [7]. The dumping of waste in landfills, industrial manufacturing, and the development of agricultural technologies are all related to the release of primary and secondary MPs. In addition, since plastic waste can have a physical and chemical effect on soil, there is an urgent need to investigate the impact of plastic on the soil.

As an entity, soil provides various services such as the nutrient cycle, carbon sequestration, and biodiversity promotion [8,9]. MPs have caused several side effects on soil functions, structure, and groundwater systems. MPs in soil contain plastic additives, heavy metals, and organic contaminants, which have had several side effects on soil density, structure, nutritional status, and the groundwater system [10,11]. Such MPs may be absorbed by plant roots or animals like earthworms and insects, even reaching humans and affecting MPs' intake rate. The main sources of plastic pollution in the soil environment are soil conditioners, mulch film, the use or overflow of contaminated irrigation water, landfills, garbage dumping, and pollution from the atmosphere. Recently, soil pollution by MPs has been related to the use of sewage sludge from wastewater treatment plants as agricultural fertilizer, which is emerging as a major source of microplastic pollution in the agricultural soil environment [12,13]. Weithmann et al. (2018) identified 146 MPs(ea)/kg (size: 1 mm to 5 mm) in dry compost based on household waste [14]. Bläsing and Amelung (2018) estimated that between 0.08 and 6.3 kg/ha of MPs would accumulate in the soil each year under the recommended compost application rate of 30 to 35 ton/ha per year [15]. Meanwhile, 40% of the total sewage sludge production in Europe and North America is used in agriculture [16]. The EU 86/278/EEC and Code 503 regulations in Europe and the United States restrict land use due to the possibility of hazardous substances in sewage sludge, but plastics are not included in the restricted (regulated) hazardous substances [17]. MPs exposed to the surface are regenerated into further MPs or nanoplastics that become reduced in size through the weathering process; they can reach the soil and groundwater environment from the soil layer to the groundwater aquifer [18]. In particular, an agricultural mulching film made of biodegradable LDPE material has a shorter biodegradation period than general plastics; therefore, it is broken into small particles, and there is a risk of soil and groundwater microplastic contamination [19].

To the best of our knowledge, research on MPs in landfill soil is ongoing but limited. Silva et al. (2021) studied MPs in landfill leachate, identified 0–291 MPs ea per liter, and reported that MPs could be reduced by certain landfill conditions and methods [20]. Mahesh et al. (2023) investigated landfill soil and found 180–1120 MPs ea per kg in the soil, confirming that polyethylene (PE) and polypropylene (PP) were the most common of the MPs [21]. As such, investigating MPs in landfill soils can help identify potential MP exposures because multiple wastes, including plastic waste, have been dumped and accumulated over decades. To understand the actual situation of MPs in Korea, this study investigated the amount, type (PP, PE, PVC, etc.), shape, and color of the MPs in the soil near the two operating landfills.

## 2. Materials and Methods

### 2.1. Study Site

Microplastics in the soil were confirmed by selecting two landfills that deal with household waste in the Gyeongnam region of Korea. Soil samples from landfills were collected from the Jinyeong landfill in Gimhae, Gyeongnam, and the Cheonseon landfill in Changwon, Gyeongnam, both in the southern part of Korea. Figure 1 shows the landfill sites and soil sample collection points of the two landfills. The Jinyeong landfill is located at 35.29′ N, 128.78′ E, and has been operating since December 1997. The landfill area is 46,989 m$^2$, and about 45 tons/day of waste is landfilled. Landfill waste is confirmed to be about 70% incinerated ash from combustible household waste and about 30% non-combustible household waste. The quantity of leachate is 120 tons/day, and connected to

the nearby wastewater treatment facility. The average annual temperature of the Jinyeong landfill in 2021 was 18 degrees, and the average annual precipitation was confirmed to be 1552.6 mm.

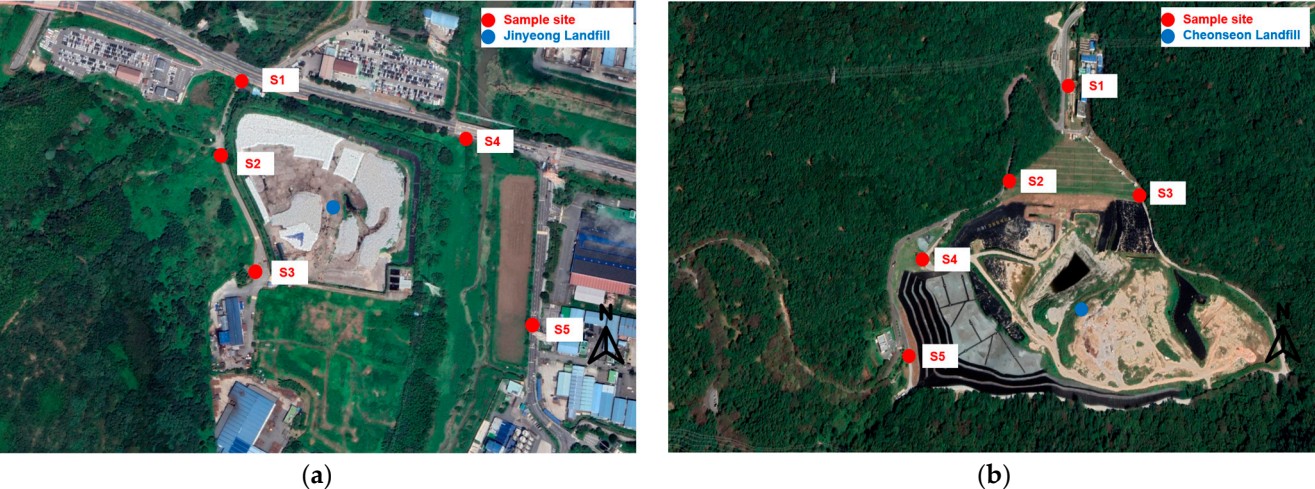

(**a**)　　　　　　　　　　　　　　　　　　　(**b**)

**Figure 1.** Sampling sites and soil collection points. (**a**) Jinyeong landfill sites and soil collection points; (**b**) Cheonseon landfill sites and soil collection points.

The Cheonseon landfill is located at 35.17′ N, 128.70′ E and has been operating since December 1993. The landfill area is 163,174 m², and about 153 tons/day of waste is landfilled. A total of 130 tons/day of leachate is connected to its own wastewater treatment facility. Landfill waste is identified as 100% non-combustible household waste. The annual average temperature of the Cheonseon landfill in 2021 was 14.8 degrees Celsius, and the average annual precipitation was 1534.1 mm.

### 2.2. Sampling and Sample Preparation

In this experiment, five samples were taken twice from the boundary of the two landfills to check MPs in the soil. All soil samples were randomly sampled twice, 500 g each, near the landfill road boundaries according to landfill regulations. Each sample was taken from 0 to 25 cm of topsoil using a stainless shovel during March 2023. In many studies, the soil sampling depth for MPs ranged from 0 to 40 cm, and sampling was performed in a single layer or multiple layers of soil from 20 cm above the surface [22]. The sample was transported to the laboratory at room temperature in a stainless steel container, spread on a stainless steel tray, and dried in an oven at 40 ± 2 °C for one day to remove moisture. The moisture content of the dried soil was calculated using Equation (1).

$$\text{Percentage of mosture contents} = \frac{\text{Wet soil weight} - \text{Dried soil weight}}{\text{Wet soil weight}} \times 100 \quad (1)$$

The sample pH was measured using a pH meter (pH700; Eutech, Vernon Hills, IL, USA). The dried soil was sieved with a stainless steel mesh using a 5 mm and 1 mm sieve. Samples that did not pass through a 1 mm sieve (Large MPs: L-MPs) and samples that passed through a 1 mm sieve (Small MPs: S-MPs) were separated. They were each stored in a 1 L glass bottle. Figure 2 shows the sample preparation for the experiment.

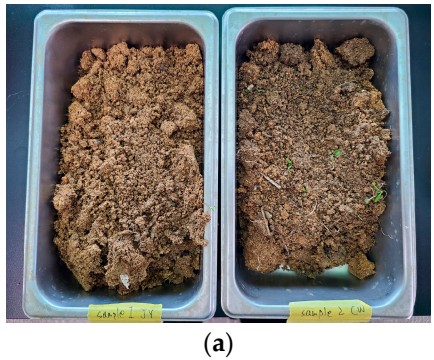
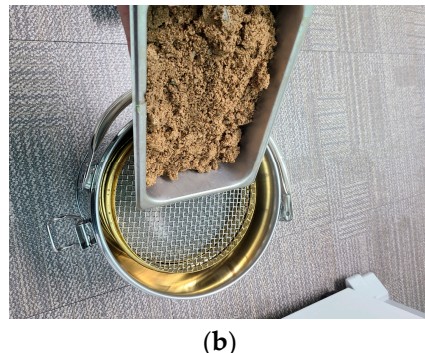

(**a**)　　　　　　　　　　　　　　　　　　　(**b**)

**Figure 2.** Soil sampling for microplastic analysis. (**a**) Picture of collected soil samples; (**b**) MPs separation by sieving.

## 2.3. Sample Pretreatment

In this study, the pretreatment process consisted of first-density separation, organic matter decomposition, second-density separation, and vacuum filtration, in that order. Density separation is a method of separating microplastic particles after suspending them using a solution having a higher density than MPs. The density of general rock is 2.65 kg/L, and the bulk density of soil is generally 1.2 kg/L or more [23]. Several studies of suspending microplastic particles using a NaCl-saturated solution with a density of about 1.2 kg/L have been reported [24–26]. The density of plastic is 0.85~0.92 kg/L for PP, 0.89~0.93 kg/L for LDPE, 0.94~0.97 kg/L for HDPE, and 1.04~1.08 kg/L for PS. The density for nylon 6 (PA6) is 1.15 kg/L, that for PVC is 1.16~1.41 kg/L, and that for PET is 1.38~1.41 kg/L [27,28]. Therefore, it was necessary to suspend the microplastic particles using an appropriate solution, and lithium metatungstate (LMT, CAS number 13762-75-9; Aladdin, Pompano Beach, FL, USA) was used as a density separation reagent in this study [29]. In a container, 1 L of 5.4 mol LMT solution (1.62 kg/L) was mixed with 1 kg of a soil sample and left in a glass beaker for 24 h. After 24 h, 200 mL of the upper part of the sample after density separation was collected by using a vacuum pump on a stainless steel mesh filter disk (pore size 20 μm, diameter 25 mm). It is important to reduce the use of saturated solutions because excessive use of saturated solutions for density separation can increase experimental costs, cause environmental contamination, and safety problems. Naji et al. (2019) applied the air-induced overflow (AIO) method, which increased the floating probability of light microplastic particles by forming an airflow of about 0.1 L/s with air to increase the efficiency of density difference separation [30].

After the first-density separation, the removal of interfering substances other than MPs was carried out. Various organic matters exist in the soil. If these are not completely decomposed, identifying MPs is difficult, and further instrumental analysis results would be imprecise. Chemicals such as $H_2O_2$, $FeSO_4$, $NaOH$, and $HNO_4$ are often used to remove organic matter, but using a chemical treatment that is too strong should be avoided to prevent MPs from melting or deteriorating [31,32]. Hence, 30% $H_2O_2$ was used to remove organic matter completely from the soil. The samples treated by first-density separation were placed into glass beakers with the reagent solution at a ratio of 1:10 (10 g of soil, 100 mL of solution). Then, the glass beaker was shaken at 40 °C, 125 rpm for 3 days.

After the organic matter removal, the second-density separation was carried out using a 5.4 mol LMT solution. After 24 h, 100 mL of the supernatant was collected. Then, the sample was filtered with a stainless filter paper, rinsed with tertiary distilled water, and dried. The filtered stainless paper was dried at 40 °C for 24 h and then used for FT-IR microscope analysis. Figure 3 shows the pretreatment steps before FT-IR microscope analysis.

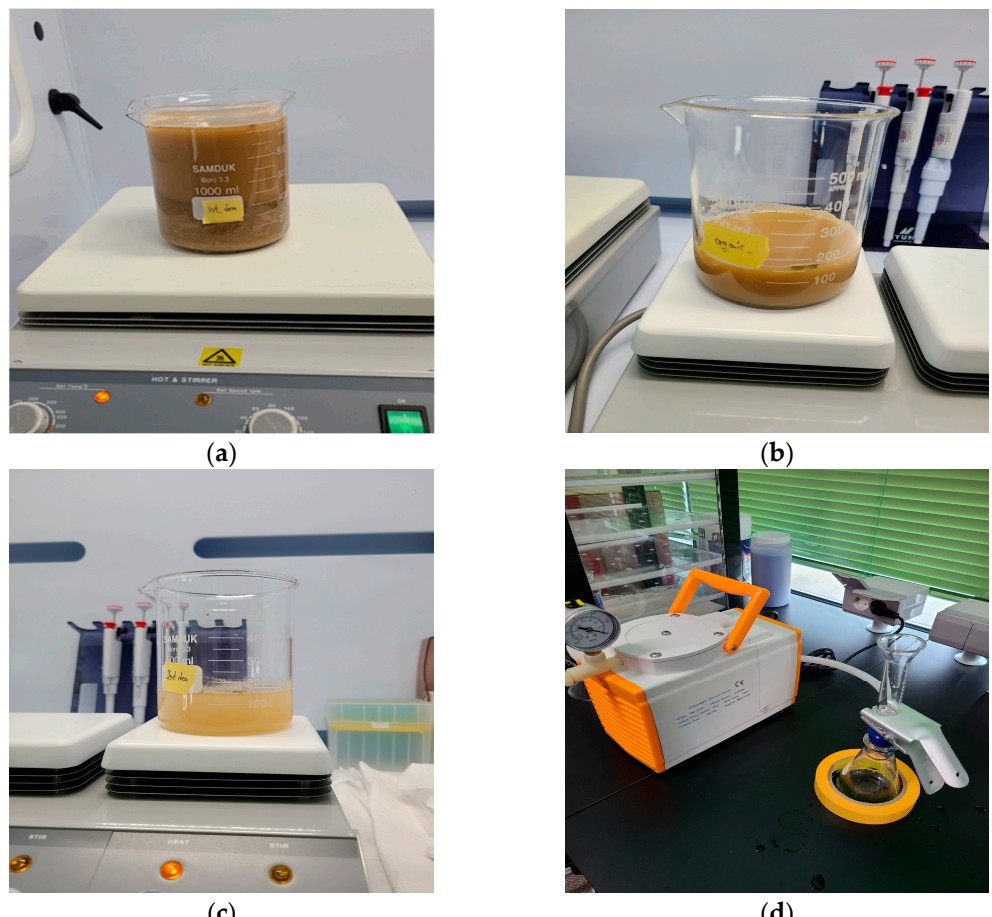

**Figure 3.** Pretreatment process for MPs' analysis. (**a**) Picture of first-density separation; (**b**) picture of organic matter decomposition; (**c**) picture of second-density separation; (**d**) picture of vacuum filtration.

*2.4. Microplastic Analysis*

The analysis of MPs to characterize their properties uses destructive or non-destructive analysis methods [33]. Destructive analysis methods include GC/MS (gas chromatography–mass spectrometry) and HPLC (high-performance liquid chromatography). Non-destructive analysis methods include Fourier transform infrared spectroscopy (FT-IR) spectroscopy, Raman spectroscopy, laser direct infrared, and the Nile red method, but FT-IR spectroscopy is the most commonly used [34]. FT-IR microscope spectroscopy enables qualitative/quantitative analysis of MPs by visualizing the filter paper through which the sample was filtered. The FT-IR microscope can check the material, size, number of MPs, polypropylene (PP), polyethylene (PE), polystyrene (PS), acrylic, polytetrafluoroethylene (PTFE), polyethylene terephthalate (PET), polyvinyl chloride (PVC), alkyd, polyurethane (PU), etc. However, the disadvantage is that it takes a lot of time and is difficult to apply to small, irregular shapes and samples of 20 μm or less.

This study also analyzed MPs' chemical characteristics using a FT-IR microscope (IN10MX; Thermo Fisher, Scientific, Waltham, MA, USA) and a digital microscope (DSZM-7045T; Dongwon Industry, Seoul, Korea). The settings of the FT-IR microscope for microplastic analysis are described in Table 1. Figure 4 shows the microplastic analysis results as obtained using the FT-IR microscope. The procedures for the FT-IR microscope analysis were to prepare a sample (mesh filter), run the FT-IR microscope (turn it on one day before use, stabilize it, and then insert liquid nitrogen before use), mount the sample on the holder, and create mapping and images in transmission mode for small plastics, and in ATR (attenuated total reflection) mode for large plastics. After the mapping was completed,

FT-IR analysis software (Omnic 9) was used to match the library to check the number, size, and properties of MPs. The checked microplastic filters were used to analyze the color and shape of the microplastic in a digital microscope. Figure 5 shows the shapes of MPs during the experiment, and Figure 6 presents the colors of MPs during the experiment. The shapes were classified into fragments, fibers, spheres, and films, and the colors were identified as yellow, white, transparent, red, green, gray, blue, and black.

**Table 1.** FT-IR microscope settings for microplastic analysis.

| Index | Contents |
|---|---|
| Collection Mode | Transmission |
| Detector | Imaging_MCT |
| Collection Time | 3 s |
| Spectra/Resolution | Normal |
| Scan Time | 3 s |
| Background | Collect background every 300 min |

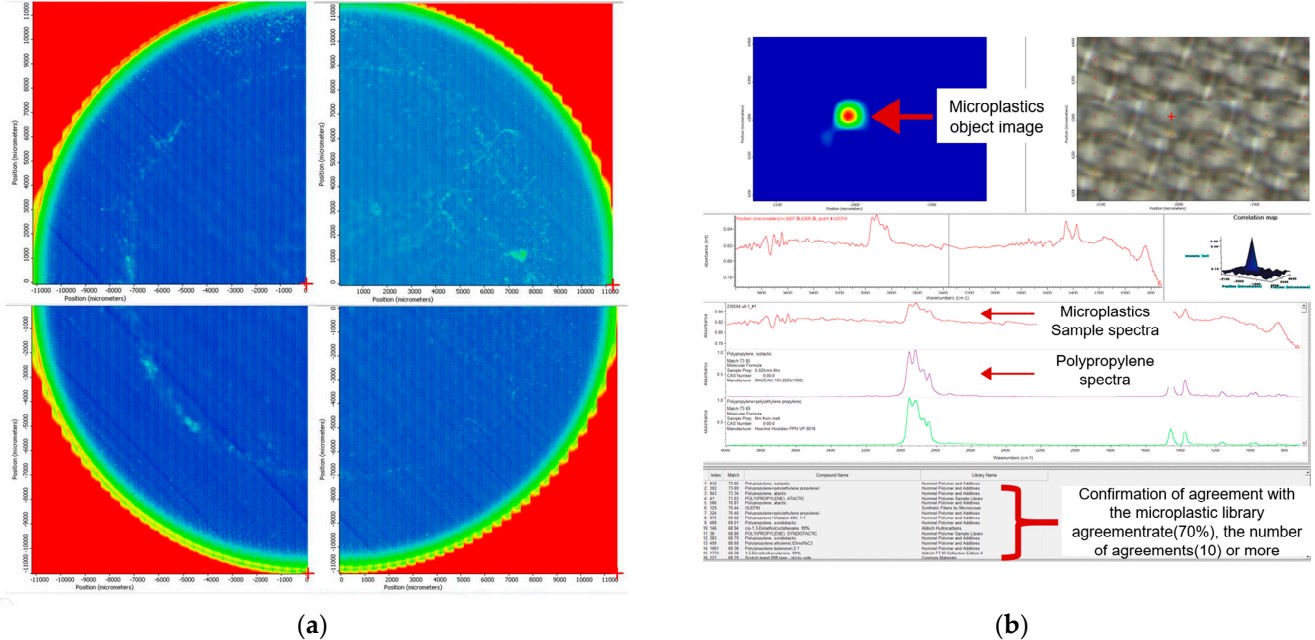

(**a**)

(**b**)

**Figure 4.** Microplastic mesh filter image and microplastic spectrum. (**a**) Microplastic mesh filter image; (**b**) microplastic spectrum.

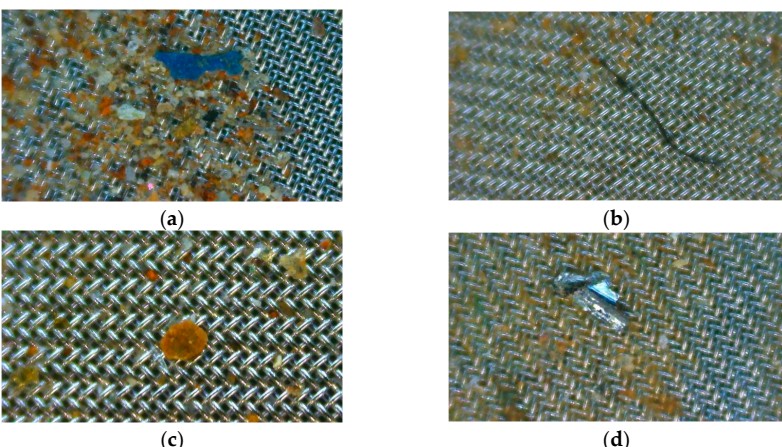

(**a**)

(**b**)

(**c**)

(**d**)

**Figure 5.** Shapes of microplastic. (**a**) Fragment; (**b**) fiber; (**c**) sphere; (**d**) film.

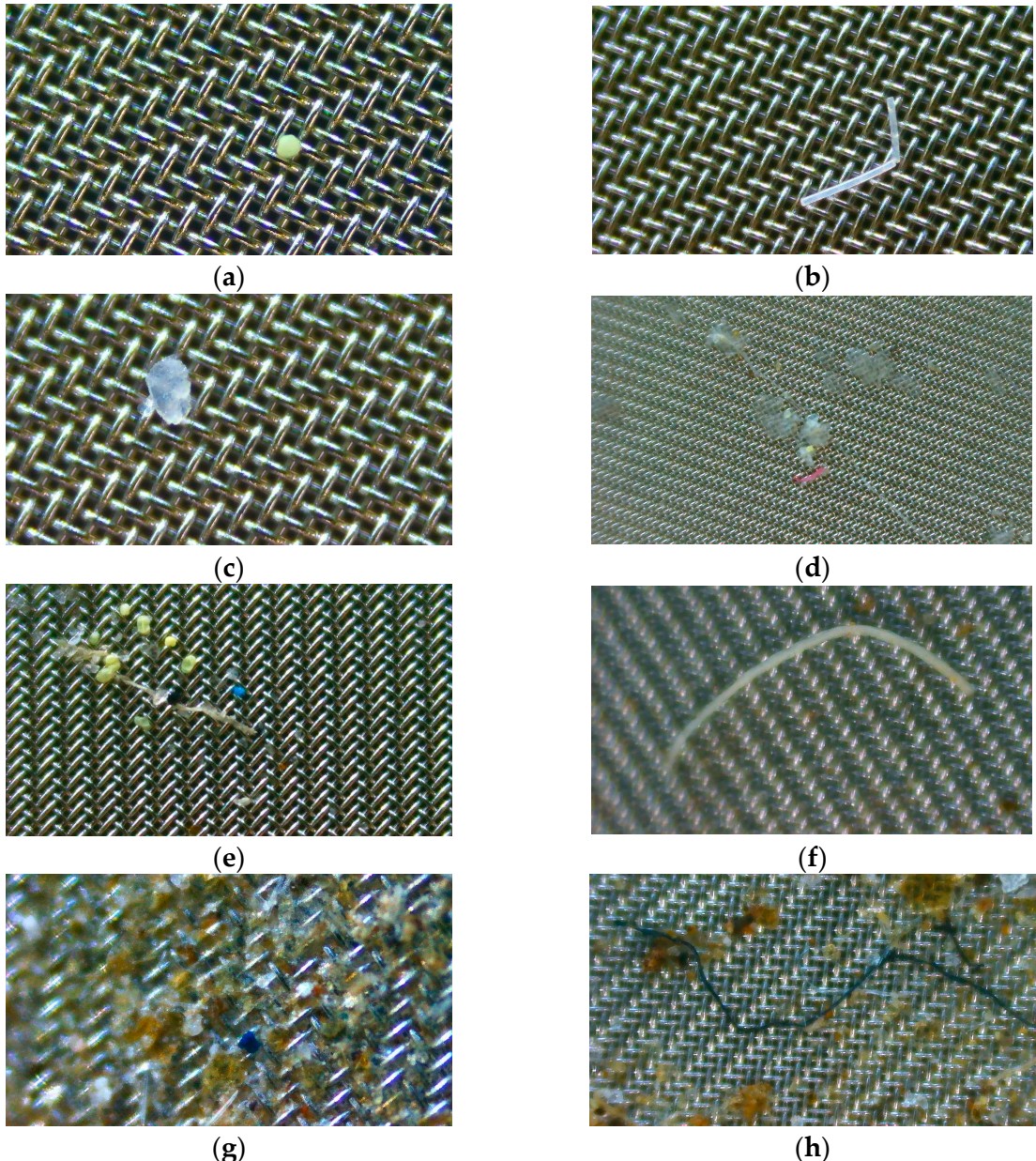

**Figure 6.** Colors of microplastic. (**a**) Yellow; (**b**) white; (**c**) transparent; (**d**) red; (**e**) green; (**f**) gray; (**g**) blue; (**h**) black.

## 3. Results

### 3.1. Soil Microplastic Contamination in the Jinyeong Landfill

Table 2 shows the moisture content and pH of the soil collected from the five locations of the waste landfill in Jinyeong, and the number of MPs in it. The moisture content of the soil samples was found to be between 10 and 22%, and the driest soil was identified in sample 3, near the landfill. The pH of the samples was confirmed to be between 5.9 and 6.9, and the soil of sample 3 had the lowest pH of 5.9.

**Table 2.** Moisture content, pH, and number of MPs in the soil.

| Category | Note | Moisture Content (%) | pH | MPs (ea)/kg | |
|---|---|---|---|---|---|
| Sample 1 | Entrance of landfill | 16 | 6.5 | L-MP | 15 |
| | | | | S-MP | 65 |
| | | | | Total | 80 |
| Sample 2 | Near landfill | 20 | 6.8 | L-MP | 8 |
| | | | | S-MP | 72 |
| | | | | Total | 80 |
| Sample 3 | Near landfill | 10 | 5.9 | L-MP | 11 |
| | | | | S-MP | 38 |
| | | | | Total | 49 |
| Sample 4 | Near landfill | 22 | 6.7 | L-MP | 8 |
| | | | | S-MP | 82 |
| | | | | Total | 90 |
| Sample 5 | Near landfill | 18 | 6.9 | L-MP | 12 |
| | | | | S-MP | 56 |
| | | | | Total | 68 |

The amount of MPs was found to be 49–90 MPs(ea)/kg. Sample 3 contained the least, with 49 ea(MPs)/kg, and sample 4 contained the most, with 90 MPs(ea)/kg. The lowest quantity of MPs was detected in sample 3, which had the least amount of moisture, but it was difficult to find a correlation between MPs, moisture, and pH. Among the MPs, the amount of L-MP (1 mm–5 mm) was relatively smaller than that of S-MP (less than 1 mm), but the correlation between the number of L-MP and S-MP could not be confirmed.

Figure 7 shows the ratio of MPs in different shapes as fragments, fibers, spheres, and films. Fibers showed the highest ratio, up to 26.4–45.0% (Avg. 36.0% + SD 7.5%), followed by films at 22.5–36.7% (Avg. 28.3% + SD 6.0%), fragments at 17.5–31.3% (Avg. 23.3% + SD 5.5%), and spheres at 6.7–17.6% (Avg. 12.4% + SD 4.2%). Although there were differences depending on the sample, it was confirmed that, among the MPs, fibers accounted for the highest proportion, followed by films and fragments.

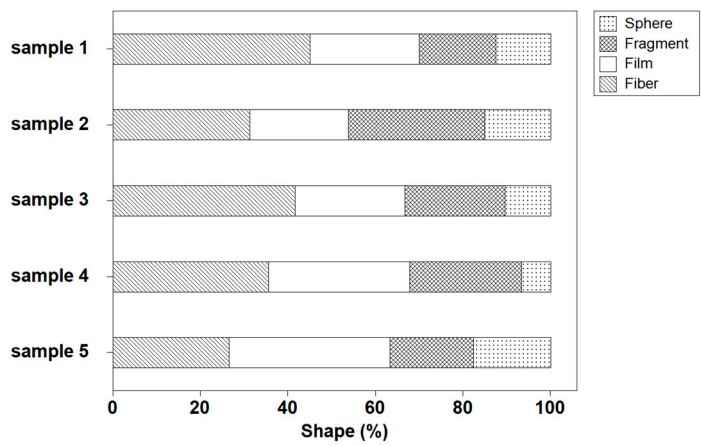

**Figure 7.** Percentages of MPs by shape in five samples.

Figure 8 shows the percentages of microplastic types in five samples. The types of MPs from sample 1 to sample 5 were confirmed as polyethylene (PE) at 26.5–37.5% (Avg. 32.3% + SD. 4.2%), polypropylene (PP) at 29.4–37.5% (Avg. 33.0% + SD 3.4%), polyethylene terephthalate (PET) at 6.3–17.6% (Avg. 13.1% + SD 4.5%), nylon at 7.5–14.7% (Avg. 10.8% + SD 2.9%), polystyrene (PS) at 2.9–6.3% (Avg. 4.5% + SD 1.3%), polyvinyl chloride (PVC) at 1.1–7.4% (Avg. 4.0% + SD 2.4%), and poly methyl methacrylate (PMMA) at 0–3.8% (Avg. 2.8% + SD 1.0%). PE and PP were detected at a similar rate of more than 65% and identified as major microplastic sources.

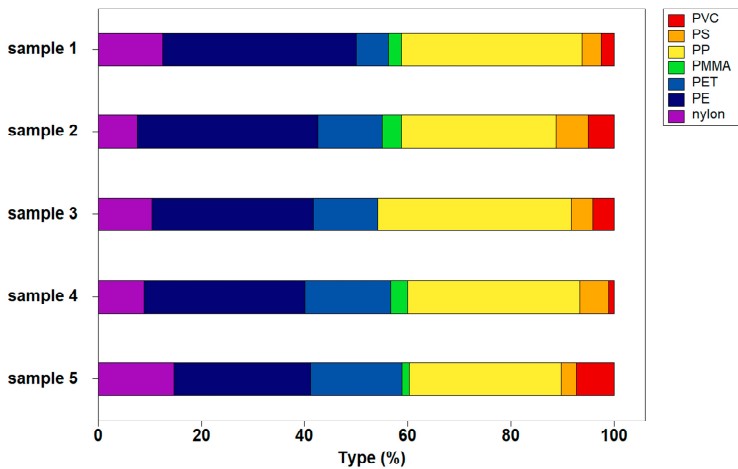

**Figure 8.** Percentages of MPs by type in five samples.

The results for each MP's color for the five samples are shown in Figure 9. As a result, black 22.9–37.5% (Avg. 29.4% + SD 7.2%), blue 11.8–25.0% (Avg. 19.3% + SD 4.8%), white 8.8–20.8% (Avg. 14.1% + SD 4.7%), gray 8.3–14.7% (Avg. 11.5% + SD 2.5%), transparent 4.4–12.5% (Avg. 8.6% + SD 3.4%), red 3.8–8.9% (Avg. 6.2% + SD 1.8%), green 3.8–8.8% (Avg. 6.2% + SD 2.2%), and yellow 2.5–6.3% (Avg. 4.7% + SD 1.4%) were confirmed sequentially. It was identified that the ratio of black was the highest, followed by blue and white.

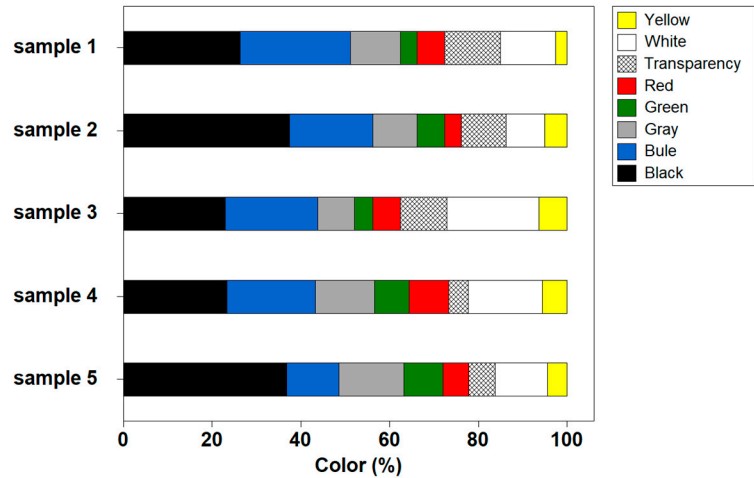

**Figure 9.** Percentages of MPs by color in five samples.

### 3.2. Soil Microplastic Contamination in the Cheonseon Landfill

Table 3 shows the water content, pH, and number of MPs in the soils of the five samples. Soils from sample 1 to sample 5 collected from the Cheonseon landfill were found to have a water content of 9 to 21% and a pH from 6.4 to 7.1. The soil with the highest moisture content was sample 5, collected from outside the landfill boundary, and the soil with the lowest moisture content was identified as sample 1, with 9% moisture. As for the pH, sample 4 was closest to neutral at 7.2, and sample 1 was the lowest at 6.4. The detected amounts of MPs were 109 MPs(ea)/kg in sample 1, 113 MPs(ea)/kg in sample 2, 82 MPs(ea)/kg in sample 3, 104 MPs(ea)/kg in sample 4, and 81 MPs(ea)/kg in sample 5. Sample 2 had the highest amounts of MPs, with 113 MPs(ea)/kg, and sample 5 had the lowest amounts, with 81 MPs(ea)/kg. The correlation between water content and pH confirmed that the dry soil had a relatively low pH. It was confirmed that MPs appeared regardless of soil properties, and there was no correlation between L-MP and S-MP.

**Table 3.** Moisture content, pH, and number of MPs in the soil.

| Category | Note | Moisture Content (%) | pH | MPs (ea)/kg | |
|---|---|---|---|---|---|
| Sample 1 | Entrance of landfill | 9 | 6.4 | L-MP | 25 |
| | | | | S-MP | 84 |
| | | | | Total | 109 |
| Sample 2 | Near landfill | 15 | 7.1 | L-MP | 21 |
| | | | | S-MP | 92 |
| | | | | Total | 113 |
| Sample 3 | Near waste drop off | 12 | 6.8 | L-MP | 20 |
| | | | | S-MP | 62 |
| | | | | Total | 82 |
| Sample 4 | Outside of landfill boundary | 18 | 7.2 | L-MP | 13 |
| | | | | S-MP | 91 |
| | | | | Total | 104 |
| Sample 5 | Outside of landfill boundary | 21 | 6.7 | L-MP | 15 |
| | | | | S-MP | 66 |
| | | | | Total | 81 |

The microplastic shape characteristics of the five samples are shown in Figure 10. Looking at the shape characteristics of MPs from sample 1 to sample 5, it was confirmed that fragments were 27.5 to 43.2% (Avg. 32.2% + SD 6.3%), fibers were 29.3 to 37.2% (Avg. 34.4% + SD 3.2%), films were 13.6 to 29.8% (Avg. 23.8% + SD 6.4%), and spheres were 6.2 to 12.2% (Avg. 9.5% + SD 2.6%). Fibers were found at the highest rate, followed by fragments, films, and spheres. It was confirmed that the shape ratio differed depending on the sample, but it was in the form of MPs with many fibers, fragments, and films.

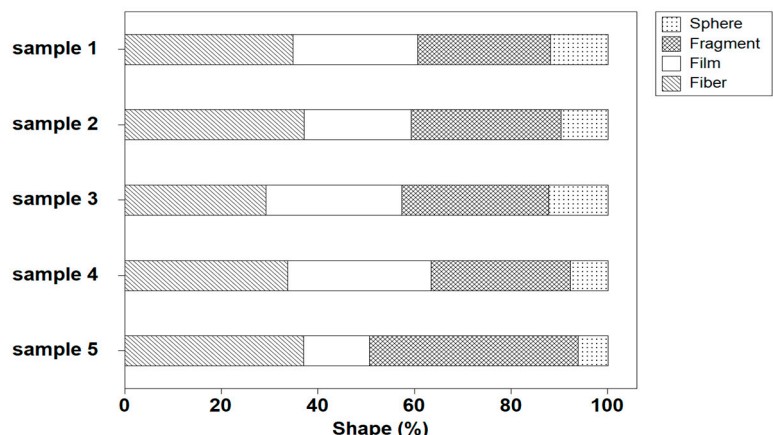

**Figure 10.** Percentages of MPs by shape in five samples.

Figure 11 shows the types of MPs by percentage for the five samples. The types of MPs from sample 1 to sample 5 were found to be as follows: PE 25.9 to 38.5% (Avg. 31.2% + SD 5.6%), PP 27.5 to 36.6% (Avg. 31.3% + SD 3.4%), PET 8.5 to 15.9% (Avg. 12.7% + SD 3.3%), PS 1.8 to 12.3% (Avg. 6.5% + SD 3.9%), nylon 3.7 to 11.1% (Avg. 7.4% + SD 3.0%), PMMA 3.7 to 6.2% (Avg. 5.4% + SD 1.9%), and PVC 4.6 to 6.2% (Avg. 5.5% + SD 0.7%). The sum of PP and PE was 62.5%, confirming that they were major sources of MPs.

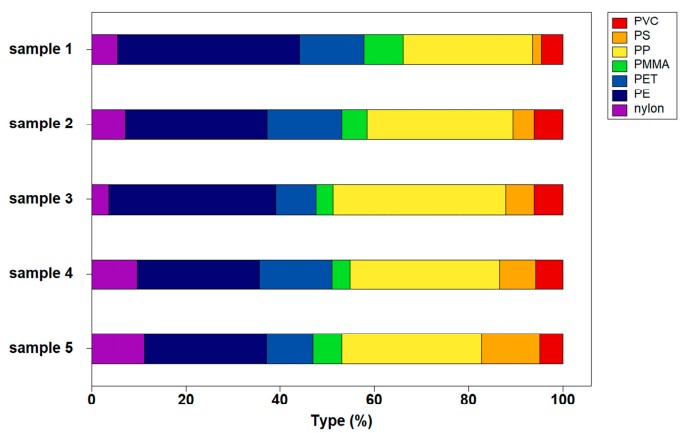

**Figure 11.** Percentages of MPs by type in five samples.

Figure 12 shows the color of MPs in percentages of the five samples. The colors of MPs were discovered to be black at 24.8 to 30.5% (Avg. 28.2% + SD 2.3%), white at 12.3 to 19.2% (Avg. 15.7% + SD 2.5%), blue at 10.6 to 16.5% (Avg. 14.2% + SD 2.2%), transparent at 6.7 to 15.9% (Avg. 10.4% + SD 3.7%), gray at 5.5 to 11.1% (Avg. 9.6% + SD 2.3%), green at 4.9 to 13.3% (Avg. 8.4% + SD 3.1%), red at 6.2 to 8.3% (Avg. 7.7% + SD 1.4%), and yellow at 3.5 to 7.7% (Avg. 5.9% + SD 1.8%). The results show that black, white, and blue were the main colors of MPs.

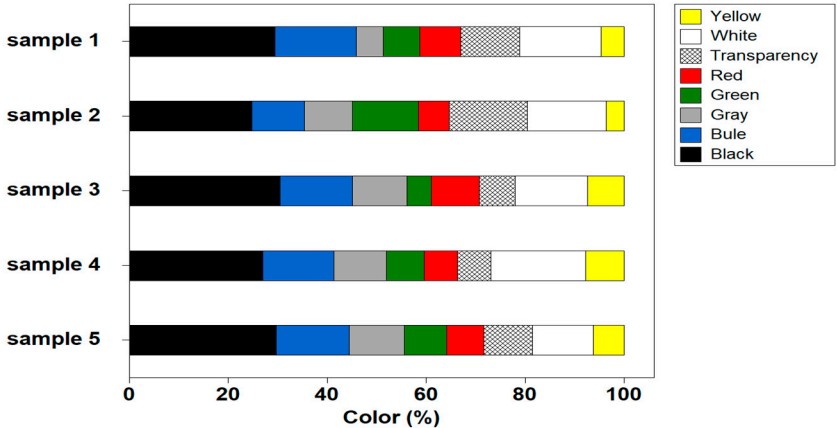

**Figure 12.** Percentages of MPs by color in five samples.

## 4. Discussion

### 4.1. Quantities and Shapes of MPs in the Soil near a Landfill

In this study, MPs were identified by sampling the soil around two landfill sites in Korea. The two sites had different levels of MPs, as an average of 73.4 MPs(ea)/kg was found in the soil near the Jinyeong landfill and an average of 97.8 MPs(ea)/kg in the soil near the Cheonsan landfill. Mahon et al. (2017) investigated MPs in the sludge of wastewater treatment plants and identified 4196–15,385 MPs(ea)/kg using scanning electron microscopy (SEM) analysis [35]. Weber et al. (2022) identified MPs in farmland and found 0.00–56.18 MPs(ea)/kg using FTIR analysis [36]. Sun et al. (2021) investigated 28 soils in six sites of farmland in China and found 1.37 to 11.51 MPs(ea)/kg using FTIR analysis [37]. Mahesh et al. (2023) identified MPs in landfill soil and found 180 to 1120 MPs (ea)/kg using FTIR analysis [21]. Liu et al. (2023) investigated the agricultural soil around the soil waste treatment center and confirmed 280 to 2360 MPs (ea)/kg using Raman spectroscopy and SEM-EDS analysis [38]. The above studies used FTIR, Raman, and SEM analysis methods, but the difference in the number of microplastics according to the analysis method would not be large because non-destructive methods and precise instruments were used. Accordingly, it was found that the results in this paper were slightly higher than the

number of MPs in general agricultural soil and slightly lower than the number of MPs in the soil around the landfill. Differences in amounts of microplastics between the Cheonsan and Jinyoung landfills can be affected by several factors, such as the type of waste, the characteristics of the soil, the environment around the landfill, and the collection sites. However, in this study, only the type of waste was identified. As for the type of waste, it was confirmed that the Cheonsan landfill dumped 100% non-combustible household waste, and the Jinyoung landfill dumped 70% incinerated ash from combustible household waste and 30% non-combustible household waste. It is confirmed that the amount of microplastics from non-combustible municipal waste is higher than that of incinerated ash from combustible household waste. However, it is necessary to define the level of MP contamination through more studies on MPs in soils near landfills.

In the two landfill sites, the shapes of MPs were mainly fragments, fibers, and films, with high sum ratios of 87.6% and 90.5% for the Jinyeong and Cheonsan landfills, respectively. As for plastic shapes in the soil, the contamination of secondary plastics (fragmented from large plastics) was higher than that of primary plastics (microplastics from the time of production). Lozano et al. (2021) reported that microplastics could affect soil aggregation and root microplastic absorption capacity [39]. Lehmann et al. (2021) also confirmed that MPs are important regulators of soil aggregation and organic matter decomposition according to their shapes, and the fiber type has a negative effect on aggregate formation [40]. As such, microplastic shapes in soil may have a significant effect on soil biodegradation and structure.

*4.2. Types and Colors of MPs in Soil near Landfills*

The MPs' type and color are important data for determining the origin of MPs [41]. This study identified PP and PE as the highest, with 65.3% in the Jinyeong landfill and 62.5% in the Cheonsan landfill. Although there was a difference in the number of MPs in the two landfills, the main plastic types were identified as being PE and PP. PE is widely used in agricultural mulch films, disposable products, pipes, toys, food containers, vinyl, and pipes and is the largest amount produced among plastics [5]. PP has the second-highest production rate, at 19.7%, among plastics and is widely used in food packaging, masks, tires, pipes, and vehicle products [5]. Other microplastic studies also confirmed that PP and PE were found in high proportions [42,43].

As for the color of MPs, it was found that black showed the highest ratio of 29.4% and 28.2% for the Jinyeong and Cheonsan landfills, respectively. Black plastics are used in everyday life for various purposes, such as tires, wires, textiles, vinyl, masks, and packaging containers. In the case of black MPs, there could be various origins, but in the case of the two landfills, one of the main causes would be tire wear because they were located next to the road. Additionally, in the case of PP, the proportion of black plastic was high. For the other colors, white, blue, clear, gray, green, red, and yellow, various plastic origins could be expected.

**5. Conclusions**

This study investigated the level of MPs contamination in the soil near two landfills in Korea. The amount of MPs in the soil was confirmed to be 73.4 MPs(ea)/kg on average for the Jinyeong landfill site and 97.8 MPs(ea)/kg for the Cheonsan landfill site. The shapes of the MPs were mainly composed of fragments, fibers, and films, and it was confirmed that secondary plastics were found at a high rate. As for the types of plastics, PP and PE were identified at a high rate of 62.5 to 65.3%, followed by PET, PS, nylon, PMMA, and PVC. As for the color of the MPs, black was identified at the highest rate, and other MPs that were white, blue, transparent, gray, green, red, and yellow were identified. The concentration of MPs near the landfill site was higher than that of general farmland and slightly lower than that of the soil near the landfill site. The results of this study can be taken as partial data on MPs in the soil around the landfill. However, more studies on MPs in the soil around the landfill sites should be investigated to understand the current situation of

MP contamination. This paper will establish potential measures for soil purification and protection from MP contamination.

**Author Contributions:** Conceptualization, G.-Y.H., W.-K.K., H.P. and K.I.; methodology, G.-Y.H. and H.P.; software, W.-K.K. and H.P.; formal analysis W.-K.K. and H.P.; investigation, W.-K.K. and H.P.; data curation, W.-K.K., H.P. and G.-Y.H.; validation, G.-Y.H.; writing—original draft preparation, W.-K.K. and H.P.; writing—review and editing, G.-Y.H. and K.I.; supervision, K.I. and G.-Y.H.; All authors have read and agreed to the published version of the manuscript.

**Funding:** This research received no external funding.

**Institutional Review Board Statement:** Not applicable.

**Informed Consent Statement:** Not applicable.

**Data Availability Statement:** The data presented in this study are available upon request from the corresponding author.

**Conflicts of Interest:** The authors declare no conflict of interest.

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
