# Peer review of "Investigation on Microplastics in Soil near Landfills in the Republic of Korea"

_sustainability, doi:10.3390/su151512057_

Round 1
Reviewer 1 Report
No information is given as to how long the landfill sites have been exploded? Do the authors have this information?
The authors state that the Jinyeong landfill accounts for 70% of the ash. The landfill is correlated with the wastewater treatment plant? Are these ashes, is sewage sludge after incineration? It would also be useful to have information about this treatment plant (technology, capacity).
Soil quality has a significant impact on microplastic accumulation. Looser-textured soils, containing high amounts of organic matter, are better at purifying water and may prevent microplastic infiltration into deeper soil layers. In contrast, soils with a compact structure and low organic matter content may favour greater microplastic retention. Do the authors have any information on the quality of the soils around the landfills analysed?
Temperature also have an important influence on the distribution of microplastic in the soil. So what are the average annual temperature and rainfall within the analysed landfills?
Table 1 . “1 Tables may have a footer.” Remove
Figure 4. figure b is a screen shot of the programme, the programme inferface should be truncated. The figure should also be enlarged for better visibility.
Table 2 remove the underline under sample 1.
“The concentration of MPs near the landfill site was higher than that of general farmland and slightly lower than that of the soil near the landfill site.” The concentration of MPs near the landfill was slightly lower than in the soil near the landfill?
The authors presented interesting research results, but the paper lacked a discussion of the results obtained. Why the microplastic content in the area around Jinyeong landfill is lower than in Cheonsan landfill.
The authors show that the concentration of microplastic is much higher than in agricultural soils. It could therefore be presented what the microplastic concentration results are in agricultural soils, citing the results of studies available in the literature, this would give a visual contrast between the contents in different locations.
Author Response
Response to Reviewer 1 Comments
Thanks for the advice on revisons to the paper
This paper has been revised according to the recommendation of revision Reviewer 1 colored in red.
1. No information is given as to how long the landfill sites have been exploded? Do the authors have this information?
=> Thanks for the comment. I have added more information about two landfills operation date and our sample collection date.
2. The authors state that the Jinyeong landfill accounts for 70% of the ash. The landfill is correlated with the wastewater treatment plant? Are these ashes, is sewage sludge after incineration? It would also be useful to have information about this treatment plant (technology, capacity).
=> Thanks for the comment. I have added more information ‘70% of incinerated ash from the combustible household waste and about 30% of non-combustible household waste.’ The leachate is only connected to the near waste treatment plant.
3. Soil quality has a significant impact on microplastic accumulation. Looser-textured soils, containing high amounts of organic matter, are better at purifying water and may prevent microplastic infiltration into deeper soil layers. In contrast, soils with a compact structure and low organic matter content may favour greater microplastic retention. Do the authors have any information on the quality of the soils around the landfills analysed?
=> Thanks for the comment. Unfortunately, we do not have information on the soil around the landfills. However, the moisture content of the collected soil is shown in Table 2.
4. Temperature also have an important influence on the distribution of microplastic in the soil. So what are the average annual temperature and rainfall within the analysed landfills?
=> Thanks for the comment. Average annual temperature and precipitation were added to Line 99 and Line 104.
5. Table 1 . “1 Tables may have a footer.” Remove
=> Thanks for the comment. The footer has been removed.
6. Figure 4. figure b is a screen shot of the programme, the programme inferface should be truncated. The figure should also be enlarged for better visibility.
=> Thanks for the comment. The figure b has been revised by the author’s comment.
7. Table 2 remove the underline under sample 1.
=> Thanks for the comment. the underline has been removed.
8. Table 2 remove the underline under sample 1.
=> Thanks for the comment. the underline has been removed.
9. Why the microplastic content in the area around Jinyeong landfill is lower than in Cheonsan landfill.
=> Thanks for the comment. we added more discussion between line 290 and line 297.
“Differences of amounts of microplastics between Cheonsan and Jinyoung landfills can be affected by several reasons such as the type of waste, the charateristics of the soil, the environment around the landfill, and the collection sites. However, in this study, the type of waste was only identified. As for the type of waste, it was confirmed that the Cheonsan Landfill dumped 100% non-combustible household waste and the Jinyoung Landfill dumped 70% incinerate ash from combustible household waste and 30% non-combustible household waste. It is confirmed that the amount of microplastics from non-combustible municipal waste is higher than that of incinerated ash from combustible household waste. ”

Reviewer 2 Report
The research, the results of which are presented in the article entitled "Investigation on microplastics in soil near landfills South Korea," is current and important. The described results have a local character but due to the significance of the issue, they can be published in the journal "Sustainability." The article is well-written, and I have no major comments. I have provided a few minor comments for the authors below:
Line 147: Firstly, H2O2 and FeSO4 are neither acid nor base; the former is hydrogen peroxide, and the latter is a salt. Secondly, there is a repetition in the sentence: "...such as H2O2, FeSO4, H2O2, and FeSO4..."
Line 166-167: To the non-destructive methods, you can add the LDIR method (Laser Direct Infrared).
Line 209: Do L-MPs reach sizes up to 500 mm (0.5 m)???!!!
Fig. 8 and 11 are not very clear. Perhaps using colors would be better?
Line 273-286: When comparing the obtained results with the findings of other studies, it is necessary to analyze whether those studies were conducted using the same analytical method.
Author Response
Response to Reviewer 2 Comments
Thanks for the advice on revisons to the paper
This paper has been revised according to the recommendation of revision Reviewer 2 colored in red.
1. Line 147: Firstly, H2O2 and FeSO4 are neither acid nor base; the former is hydrogen peroxide, and the latter is a salt. Secondly, there is a repetition in the sentence: "...such as H2O2, FeSO4, H2O2, and FeSO4..."
=> Thanks for the comment. The sentence has been revised as “The chemicals such as H2O2, FeSO4, NaOH and HNO4 are often used for the removal of organic matter but, use of too strong chemical treatment should be avoided to prevent MPs from melting or deteriorating[31, 32]. ”
2. Line 166-167: To the non-destructive methods, you can add the LDIR method (Laser Direct Infrared).
=> Thanks for the comment. The sentence has been revised as “The chemicals such as H2O2, FeSO4, NaOH and HNO4 are often used for the removal of organic matter but, use of too strong chemical treatment should be avoided to prevent MPs from melting or deteriorating[31, 32]. ”
3. Line 209: Do L-MPs reach sizes up to 500 mm (0.5 m)???!!!
=> Thanks for the comment. We revised 500 mm to 5mm.
4. Fig. 8 and 11 are not very clear. Perhaps using colors would be better?
=> Thanks for the comment. We have changed them to using colors.
5. Line 273-286: When comparing the obtained results with the findings of other studies, it is necessary to analyze whether those studies were conducted using the same analytical method.
=> Thanks for the comment. We added analytical methods in each reference and added that the reason for using the reference data. “The analysis methods of the above papers used FTIR, Raman, and SEM analysis methods, but the difference in the amount of microplastics according to the analysis method would not be large because non-destructive methods and precise instruments were used.”

Reviewer 3 Report
Manuscript ID sustainability-2534789 Type Article Title Investigation on microplastics in soil near landfills South Korea The manuscript is well written and the data are enough to be published. I just like to see a table summarizing previous studies in this subject so the readers can compare the results. Also, if you add a paragraph in the abstract emphasizing the novelty of your work. Conclusion needs to be re written.English is ok.
Author Response
Response to Reviewer 3 Comments
This paper has been revised according to the recommendation of revision Reviewer 3 colored in red.
1. I just like to see a table summarizing previous studies in this subject so the readers can compare the results.
=> Thanks for the comment. I am sorry for not providing information of the previous study. This is our first soil microplastic investigation near landfills and the project started since January, 2023 and still going on. The next paper will contain more results in landfills and near environments.
Round 2
Reviewer 1 Report
The authors complied with the comment and answered my questions to my full satisfaction.
Congratulations